# p16 Immunohistochemical Expression as a Surrogate Assessment of CDKN2A Alteration in Gliomas Leading to Prognostic Significances

**DOI:** 10.3390/cancers15051512

**Published:** 2023-02-28

**Authors:** Lucas Geyer, Thibaut Wolf, Marie-Pierre Chenard, Helene Cebula, Roland Schott, Georges Noel, Eric Guerin, Erwan Pencreach, Damien Reita, Natacha Entz-Werlé, Benoît Lhermitte

**Affiliations:** 1Pathology Department, University Hospital of Strasbourg, 67098 Strasbourg, France; 2UMR CNRS 7021, Laboratory Bioimaging and Pathologies, Tumoral Signaling and Therapeutic Targets, Faculty of Pharmacy, 67401 Illkirch, France; 3Centre de Ressources Biologiques, University Hospital of Strasbourg, 67098 Strasbourg, France; 4Neurosurgery Department, University Hospital of Strasbourg, 67098 Strasbourg, France; 5Oncology Department, ICANS, University of Strasbourg, 67098 Strasbourg, France; 6Radiotherapy Department, ICANS, University of Strasbourg, 67098 Strasbourg, France; 7Oncobiology Platform, Laboratory of Biochemistry, University Hospital of Strasbourg, 67098 Strasbourg, France; 8Pediatric Onco-Hematology Unit, University Hospital of Strasbourg, 67098 Strasbourg, France

**Keywords:** p16, immunohistochemistry, *CDKN2A* deletion, FISH, gliomas, prognosis

## Abstract

**Simple Summary:**

The 2021 WHO classification of central nervous system tumors is a histomolecular classification system that takes into account numerous molecular data in order to better stratify patient prognoses and treatments. *CDKN2A* homozygous deletion appears to be associated with poor prognosis in many types of gliomas. The search for this deletion requires complex techniques. As *CDKN2A* encodes the p16 protein, a reliable, reproducible, and clinically meaningful IHC stain would be useful as a surrogate test. This study attempts to describe the clinical impacts of p16 immunohistochemical expression in a wide variety of gliomas, as well as its correlation with *CDKN2A* homozygous deletion.

**Abstract:**

*CDKN2A* is a tumor suppressor gene encoding the p16 protein, a key regulator of the cell cycle. *CDKN2A* homozygous deletion is a central prognostic factor for numerous tumors and can be detected by several techniques. This study aims to evaluate the extent to which immunohistochemical levels of p16 expression may provide information about *CDKN2A* deletion. A retrospective study was conducted in 173 gliomas of all types, using p16 IHC and *CDKN2A* fluorescent in situ hybridization. Survival analyses were performed to assess the prognostic impact of p16 expression and *CDKN2A* deletion on patient outcomes. Three patterns of p16 expression were observed: absence of expression, focal expression, and overexpression. Absence of p16 expression was correlated with worse outcomes. p16 overexpression was associated with better prognoses in MAPK-induced tumors, but with worse survival in *IDH*-wt glioblastomas. *CDKN2A* homozygous deletion predicted worse outcomes in the overall patient population, particularly in *IDH*-mutant 1p/19q oligodendrogliomas (grade 3). Finally, we observed a significant correlation between p16 immunohistochemical loss of expression and *CDKN2A* homozygosity. IHC has strong sensitivity and high negative predictive value, suggesting that p16 IHC might be a pertinent test to detect cases most likely harboring *CDKN2A* homozygous deletion.

## 1. Introduction

The Cyclin-Dependent Kinase Inhibitor 2A (*CDKN2A*) gene is a tumor suppressor gene located at chromosome 9, locus p21.3 that encodes the p14^ARF^ and p16^INK4a^ proteins [1]. Both proteins are generated by alternative mRNA splicing of the *CDKN2A* gene. The p16 protein is a key regulator of the cell cycle in the p16^INK4a^/cyclin D1/CDK4/Rb/E2F1 pathway and interacts closely with p53/p21/Rb signaling [2]. Both pathways are involved in various steps of the G1-S phase transition [3]. p16^INK4a^ acts by binding the Cyclin-Dependent Kinase 4/6, which inhibits its kinase activity and therefore prevents Rb phosphorylation. Rb stays bound to the transcription factor E2F1, keeping this complex within the cytoplasm and preventing E2F1 transcription of target genes involved in the G1-S phase transition [4].

*CDKN2A* is the second most impaired tumor suppressor gene in cancers, including gliomas [5], and its inactivation promotes the G1-S phase transition. The most frequently encountered molecular inactivation mechanism is *CDKN2A* homozygous deletion, but molecular inactivation can also be achieved by promoter hypermethylation, missense mutations, or post-transcriptional regulation mechanisms [6,7,8].

The prognostic impact of *CDKN2A* homozygous deletion is crucial, especially in *IDH*-mutant astrocytomas [9,10]. Thus, the last WHO classification of central nervous system tumors integrated this deletion into the grading of this type of glioma. Indeed, the presence of a *CDKN2A* homozygous deletion would now classify an *IDH*-mutant astrocytoma as WHO grade 4, even in the absence of microvascular proliferation or necrosis [11]. *CDKN2A* homozygous deletion can be detected by various techniques, such as Fluorescence In Situ Hybridization (FISH), Comparative Genomic Hybridization array (CGH-array), or potentially Next Generation Sequencing (NGS). All these methods are complex, time-consuming, and may result in delayed diagnosis and subsequent treatment. IHC is currently not recommended as a reliable surrogate test. However, there is a p16-protein-targeting antibody, which is used in daily practice for the diagnosis of HPV-induced neoplasia. Therefore, it might be of interest to evaluate IHC levels of p16 expression to provide information about *CDKN2A* alterations in other cancers. Few studies focus on the association between p16 IHC and *CDKN2A* deletion in glioma samples [12,13,14,15], but, with respect to *IDH*-mutant astrocytomas, some have reported a significant association between p16-negative IHC and a high negative predictive value. Such results promote this technique as a pertinent test to evaluate p16 expression and glioma prognosis [12,13]. Nevertheless, other publications conclude a poor correlation between *CDKN2A* results generated by FISH and IHC [14,15]. These conflicting results may be related in part to small sample size [14,15] or to cohorts comprising heterogenous groups of gliomas that did not reflect the current histomolecular classification diversity [12,13].

Here, we describe the correlation between p16 IHC expression and *CDKN2A* heterozygous and homozygous deletion assessed by FISH in a cohort of low- and high-grade diffuse pediatric- and adult-type gliomas, as well as in circumscribed gliomas. Survival analyses were performed to evaluate and compare the prognostic impacts of p16 IHC expression and the paired *CDKN2A* homozygous deletion.

## 2. Material and Methods

### 2.1. Patient Cohort

A retrospective study was conducted including one hundred and seventy-three gliomas of all grades diagnosed over a period of 13 years (1999–2012), resected in the neurosurgical department of Strasbourg University Hospital. Most of the cases were from 2005 to 2012, especially adult-type diffuse gliomas. The period of inclusion was only extended for rarer tumors, especially BRAF-altered tumors. Most cases from biopsies with limited FFPE residual samples were excluded due to the impossibility of performing tissue microarray (TMA) and additional molecular tests. Cases without informed consent and clinical follow-up were also excluded. Therefore, our cohort represents about 35% of all the patients operated on in the neurosurgical department of Strasbourg University Hospital. 

TMAs comprising 3 representative 1 mm diameter tumoral cores of all patients’ specimens were constructed. All specimens were obtained after informed consent from parents and patients and were anonymized for their analyses. This study was conducted in accordance with the ethical committee approval (declaration number: DC-2017-3090). All hematoxylin and eosin (HE)-stained slides of the entire glioma population were reviewed to reclassify all gliomas according to the 2021 WHO classification of central nervous system tumors—which also recommends IHC, FISH and/or NGS assays for an accurate histomolecular diagnosis. Supplementary IHC for the whole cohort included a search for ATRX loss of expression (ATRX antibody, clone BSB-108), p53 expression (p53 antibody, clone D07), and *IDH1 R132H* expression (*IDH1 R132H* antibody, clone IHC 132). In 50 cases for which morphological and immunohistochemical profiles were suggestive of oligodendroglioma diagnosis, the 1p/19q codeletion was detected with a FISH technique *(ZytoLight SPEC 1p36/1q25 Dual Color Probe et ZytoLight SPEC 19q13/19p13 Dual Color Probe)*. NGS was performed on diffuse high-grade pediatric gliomas and on circumscribed gliomas (68 cases), using *Illumina MiSeq* sequencing technology. Illumina NGS workflows include 4 basic steps: library preparation using the Multiplicom *Tumor Hotspot MASTR Plus kit (MR-0200.024)*, cluster generation, sequencing, and data analysis. The bioinformatics was performed using the STARK analysis environment (version 0.9b). Each technique in each case was interpreted based on the percentage of tumor cells.

### 2.2. Clinical Characteristics

Clinical data focused on patient gender, age at diagnosis, type of specimen (biopsy or surgical resection), presence of tumor recurrence, date of progression, and/or date of death. The overall survival (OS) time was calculated from the date of the initial surgery to the date of death. 

### 2.3. Evaluation of p16 and Rb1 Immunohistochemistry 

IHC was performed to evaluate p16 expression using an antibody against the p16 protein (clone E6H4). The average percentage of tumor cells exhibiting both nuclear and cytoplasmic expression was evaluated on the 3 TMA cores. Loss of Rb1 expression was assessed using an Rb1 antibody (G3-245 clone). Loss of Rb1 expression was only considered when positive control was present on the core, consisting of a positive staining of endothelial cells and micro-environmental lymphocytes. 

### 2.4. FISH Analysis

*CDKN2A* deletion status was assessed by FISH, using a dual-color FISH assay on paraffin-embedded sections with the *ZytoLight SPEC CDKN2A*/*CEN 9 Dual Color Probe* FISH probes. One hundred nuclei were evaluated to assess normal signals and the number of heterozygous or homozygous deleted tumor cells. The percentage of heterozygous or homozygous cells was related to the percentage of tumor cells evaluated on HE slides. As recommended in previous studies, a 15% cutoff of nuclei with homozygous or heterozygous deletion was set up [16].

### 2.5. Statistical Analyses

The analyses were performed using statistical websites: https://biostatgv.sentiweb.fr/ and www.pvalue.io (accessed on 7 June 2022). Qualitative variables were described as percentages and quantitative variables were described as their medians and ranges. Correlations between two groups were analyzed using Fisher’s exact test. Quantitative data were compared using Student’s t-test. All survival correlations were estimated by the Kaplan–Meier method. The log-rank test was used for univariate analyses and the Cox regression model for multivariate analyses. Variables showing prognostic significance with a *p*-value under 0.05 in univariate analysis (tumor recurrence and age over 36.5 years) were included in the multivariate analysis model. p-values under 0.05 were considered statistically significant. The ROC analysis curve was determined after calculating the True Positive Rate (TPR) and the False Positive Rate (FPR) values for several percentage thresholds of p16-positive tumor cells in the IHC assay (0%, 1%, 2%, 5%, 10%, 15%, 20%, 30%, 40%, 50%, 60%, 70%, 80%, 90%, and 100%). The TPR and FPR values were then reported on a curve using Microsoft Excel software.

## 3. Results

### 3.1. Characteristics of the Study Population

The median age of the study population was 36.5 years (range: 1 to 79 years), and 95 patients (55%) were men. One hundred and sixty-three patients (94%) had a surgical resection of the tumor, and 10 patients (6%) had a simple biopsy. Thirty-three patients (19%) were reoperated on for tumor recurrence. Table 1 summarizes the clinical information. For survival data, the median follow-up time was 62 months (range 0 to 273 months). During this follow-up period, 114 patients (66%) died. The diagnoses according to the last 2021 WHO classification of central nervous system tumors were as follows:-adult-type diffuse gliomas, including 63 *IDH*-wt glioblastomas (GBM) (considered WHO grade 4 tumors), 15 *IDH*-mutant astrocytomas (four grade 2, five grade 3, and six grade 4 cases), and 27 *IDH*-mutant 1p/19q codeleted oligodendrogliomas (OG) (10 grade 2 and 17 grade 3 cases)-18 high-grade pediatric-type diffuse gliomas, including nine H3K27-altered gliomas, one hemispheric H3.3 G34-mutant glioma, and eight wt hemispheric gliomas-47 circumscribed gliomas, including 36 pilocytic astrocytomas (PA) (considered WHO grade 1 tumors), 10 gangliogliomas (GGLs) (grade 1), and one pleomorphic xanthoastrocytoma (PXA) (with anaplastic features of WHO grade 3)-three low-grade pediatric gliomas including two subependymal giant-cell astrocytomas (WHO grade 1) and one angiocentric glioma (also grade 1)

### 3.2. Patterns of Immunohistochemical p16 Expression and Prognostic Implications

Three patterns of p16 expression were identified: (1) a total loss of expression (Figure 1A), (2) a focal positive expression in a variable amount of tumor cells (Figure 1B), and (3) an intense and diffuse positivity throughout samples (Figure 1C). These patterns were preferentially observed in different glioma types. Indeed, an absence of p16 expression was encountered in 32% of the whole cohort (56/173) and was almost restricted to high-grade glioma subtypes. Loss of p16 expression was observed in 57% (36/63) of *IDH*-wt GBM, 33% (2/6) of *IDH*-mutant astrocytomas (grade 4), 40% (2/5) of *IDH*-mutant astrocytomas (grade 3), 29% (5/17) of *IDH*-mutant 1p/19q codeleted OG (grade 3), 67% (6/9) of H3K27-altered diffuse gliomas, 38% (3/8) of wt pediatric diffuse gliomas, and in the only case of PXA with anaplasia. Among the low-grade gliomas, only one PA showed an absence of p16 expression.

A focal expression of p16 was encountered in 51% of the whole cohort (89/173). This pattern was mostly observed in low-grade MAPK-altered gliomas, especially in 83% (30/36) of PAs and 50% (5/10) of GGLs. This was also the most represented pattern of expression in low-grade adult-type gliomas, seen in 100% (4/4) of *IDH*-mutant astrocytomas (grade 2) and 100% (10/10) of *IDH*-mutant 1p/19q codeleted OG (grade 2). A subset of high-grade gliomas also exhibited this pattern: 27% (17/63) of *IDH*-wt GBM, 50% (3/6) of *IDH*-mutant astrocytomas (grade 4), 40% (2/5) of *IDH*-mutant astrocytomas (grade 3), 59% (10/17) of *IDH*-mutant 1p/19q codeleted OG (grade 3), 11% (1/9) of H3K27-altered midline gliomas, 38% (3/8) of wt high-grade pediatric gliomas, and in the only case of hemispheric G34-mutant high-grade pediatric glioma. 

An intense and diffuse expression of p16 was observed in two distinct groups of gliomas: MAPK-induced low-grade gliomas and diffuse high-grade gliomas. Indeed, this pattern was observed in 14% (5/36) of PAs and 50% (5/10) of GGLs. Among the high-grade gliomas, it was noticed in 16% (10/63) of *IDH*-wt GBM, 17% (1/6) of *IDH*-mutant astrocytomas (grade 4), 20% (1/5) of *IDH*-mutant astrocytomas (grade 3), 12% (2/17) of *IDH*-mutant 1p/19q OG (grade 3), 22% (2/9) of midline H3K27-altered gliomas, and 25% (2/8) of wt diffuse pediatric gliomas. 

Among high-grade gliomas with intense and diffuse p16 expression, all except one showed a loss of Rb1 expression. Contrarily, no case harbored a loss of Rb1 in low-grade MAPK-induced gliomas. This pattern was not encountered in other glioma subtypes. Results are summarized in Table 2. 

In the whole cohort, absence of p16 expression was significantly associated with a shorter OS in univariate analysis (Kaplan–Meier and log-rang test, *p* < 0.001). Worse outcomes were especially noticed in *IDH*-wt GBM (*p* < 0.001) (Figure 2A) and *IDH*-mutant 1p/19q oligodendrogliomas (grade 3) (*p* = 0.002) (Figure 2B). In *IDH*-wt GBM, intense and diffuse p16 expression was associated with worse outcomes when compared to focal p16 expression (Figure 2A). Nevertheless, intense and diffuse expression of p16 was associated with different prognostic impacts depending on the molecular alteration driving the tumors. In high-grade gliomas, especially in *IDH*-wt GBM, this overexpression was associated with dismal prognoses (*p* < 0.001) (Figure 2C). However, in MAPK-induced gliomas, p16 overexpression was associated with better prognoses (*p* = 0.04), (Figure 2D).

### 3.3. CDKN2A Status and Prognostic Implications

Thirty-eight *CDKN2A* homozygous deletions and 15 *CDKN2A* heterozygous deletions were identified. Homozygous deletions were essentially restricted to high-grade gliomas including 29% (5/17) of *IDH*-mutant 1p/19q codeleted OG (grade 3), 46% (29/63) of *IDH*-wt GBM, and 33% (3/9) of midline H3K27-altered gliomas. The latter was the only diffuse high-grade pediatric-type glioma to harbor homozygous deletion. No *CDKN2A* homozygous deletion was identified in grade 2 gliomas. Surprisingly, one PA presented a homozygous deletion. Heterozygous deletions were observed in high-grade gliomas including 35% (6/17) of *IDH*-mutant 1p/19q oligodendrogliomas (grade 3), 20% (1/5) of *IDH*-mutant astrocytomas (grade 3), 5% (3/63) of *IDH*-wt GBM, and 55% (5/9) of high-grade diffuse pediatric gliomas, especially H3 and *IDH*-wt. Statistically, *CDKN2A* homozygous deletion was associated with adverse prognoses in the whole population in univariate analysis (Kaplan–Meier and log-rang test, *p* < 0.001). In *IDH*-wt GBM (Figure 3A), as well as in *IDH*-mutant 1p/19q OG (grade 3) (Figure 3C), the poor prognostic outcome was also significant (*p* < 0.001).

In multivariate analyses, the correlation between *CDKN2A* homozygous deletion and worse outcomes was significant and persisted even after adjusting the population for tumor recurrence and age over 36.5 years, which are recognized as poor prognostic factors (*p* < 0.001, hazards ratio [HR]: 2.536, 95% confidence interval [CI]: 1.195-5.271 for grade 3 OG and *p* = 0.02, HR: 1.723, 95% CI: 1.012-2.936 for *IDH*-wt GBM). 

The prognostic impact in pediatric-type high-grade gliomas was not statistically significant. The only pilocytic astrocytoma with *CDKN2A* homozygous deletion showed a pejorative evolution with an OS of 60 months. 

No prognostic impact of heterozygous deletion was observed, either in the whole cohort or in precise subtypes. 

### 3.4. Comparison between p16 Immunohistochemistry and CDKN2A FISH Results

The association between *CDKN2A* homozygous deletion and the absence of p16 expression was clearly significant (*p* < 0.001). Sixty-eight percent (38/56) of gliomas with an absence of p16 expression demonstrated a *CDKN2A* homozygous deletion on FISH assays. On the other hand, none of the gliomas with intense and diffuse p16 expression (0/28) demonstrated a *CDKN2A* homozygous deletion. In the pattern defined as focal expression, there was no *CDKN2A* homozygous deletion (0/89) if more than 5% of tumor cells were expressing p16. On the ROC analysis plot, a threshold of 5% of p16-positive cells on the IHC assay showed the best performance values. Indeed, observation of less than 5% p16-positive tumor cells predicted a homozygous deletion of *CDKN2A* with a sensibility of 100%, a specificity of 71%, a positive predictive value of 50%, and a negative predictive value of 100% (Figure 3E). Survival curves of p16 pattern expression and *CDKN2A* FISH showed similar prognostic statistics (Figure 3B–E). Furthermore, survival curves of cases with absence of p16 expression with or without *CDKN2A* deletion were similar (F). 

## 4. Discussion

This study demonstrates similar prognostic values for p16 IHC absence of expression and *CDKN2A* homozygous deletion in a molecularly defined cohort comprising almost all glioma types. To obtain a significant and statistically valuable comparison, we first studied the patterns of p16 expression in the entire cohort. Three types of patterns were identified: (1) absence of expression, (2) focal expression in a variable amount of tumor cells, and (3) intense and diffuse expression across all tumor cells, which might also be considered overexpression. Interestingly, these patterns were already described in the study by Park et al. in adult-type diffuse gliomas [13], highlighting the reproducibility of these patterns and their potential extension to all glioma subtypes. As for our study, each pattern was preferentially observed in specific glioma subtypes and we were able to confirm that the absence of p16 expression was significantly correlated with worse outcomes in all glioma samples, including *IDH*-mutant gliomas. We observed that the absence of p16 expression was significantly associated with worse outcomes in *IDH*-mutant 1p/19q OG (grade 3) and in *IDH*-wt GBM. As our cohort comprised circumscribed and pediatric-type gliomas, we were able to demonstrate that p16 was focally expressed in pediatric-type, low-grade gliomas, and that overexpression of p16 was present in both *IDH*-wt glioblastomas and MAPK-induced, low-grade gliomas. 

We were able to demonstrate that the prognostic significance of p16 overexpression depended on the glioma type and, more exactly, on the molecular alteration responsible for this overexpression. In low-grade gliomas, p16 overexpression is probably related to an oncogene-induced senescence (OIS) phenomenon, which is an antiproliferative response resulting from the activation of the MAPK pathway in the presence of a functional p16 protein. These proteic features partly explain the relatively indolent course of MAPK-induced tumors, especially in low-grade and pediatric settings. In fact, the massive activation of the MAPK pathway leads to an initial hyperproliferative phase usually associated with altered DNA replication, which is followed by a DNA damage response via the p53 tumor suppressor signaling pathway. Finally, this molecular cascade triggers cellular senescence, the maintenance of which depends on the p16 pathway (Figure 4A) [4,17,18]. In contrast, in adult high-grade gliomas, especially in *IDH*-wt glioblastomas, p16 overexpression was associated with worse prognoses. In this high-grade setting, p16 overexpression results from positive feedback triggered by a loss of Rb1 expression that typically limits the G1-S transition. The loss of Rb1 expression, present in all our cases except one, is probably linked to Rb1 deletion, which has already been described in this high-grade subtype and is related to poor prognosis in some glioma subtypes [19,20,21]. p16 overexpression is, then, the witness of this molecular Rb1 alteration (Figure 4B).

Regarding the *CDKN2A* FISH results, homozygous deletions were mostly found in *IDH*-wt GBM, *IDH*-mutant 1p/19q OG (grade 3), and H3K27-altered gliomas. No homozygous deletions were identified in *IDH*-mutant high-grade astrocytomas. The prognostic impact of *CDKN2A* homozygous deletion in diffuse *IDH*-mutant astrocytomas is still under debate in the literature [9] for grade 4 subsets, but not for grades 2 and 3 [10]. Due to an insufficient number of *IDH*-mutant astrocytomas in our cohort, we could not reliably assess the prognostic impact of *CDKN2A* homozygous deletion in this grade 4 histomolecular entity. For another subtype, the *IDH*-mutant 1p/19q codeleted OG (grade 3), the presence of *CDKN2A* homozygous deletion led to very poor prognoses, close to those described in *IDH*-wt GBM. Up to now, this correlation was only underlined in the study by Appay et al. [9]. This result must be confirmed in greater cohorts; if confirmed, it would be possible to refine the WHO grading of those oligodendrogliomas harboring a *CDKN2A* homozygous deletion and/or a p16 loss of protein expression. Future investigations should focus on an unexpected grade 4 subset of oligodendrogliomas, and a *CDKN2A*-deletion or p16-protein-loss assessment must be proposed and used routinely to define those specific aggressive cases. Regarding the H3K27-altered gliomas, no significant prognostic impact was observed, probably due to the small size of our population. Nevertheless, *CDKN2A* deletion and p16 expression have been recorded, if rarely, in this pediatric population of midline diffuse gliomas, as well as in sus-tentorial high-grade tumors [22]. Our results showing frequent loss of p16 expression in the entire tumor or focally (7/9 H3 K27M and 6/8 wt cases) have implications that should be considered more largely, as they may open the path for therapies targeting the p16 pathway (e.g., CDK4 or cyclin D1) in this population. 

In low-grade gliomas, *CDKN2A* homozygous deletion seems to be an extremely rare event. No low-grade adult-type diffuse gliomas presented such deletion in our cohort. Our results therefore suggest that evaluating *CDKN2A* status in adult low-grade diffuse gliomas is not mandatory. Nevertheless, although *CDKN2A* homozygous deletion was rare in our cohort, with only one case of PA, its presence in cases of low-grade MAPK-induced gliomas showed a dramatic clinical evolution, pointing out the possibly strong prognostic impact of *CDKN2A* homozygous deletion in those populations [23,24]. *CDKN2A* heterozygous deletion was observed in many tumor types without any prognostic impact.

Nevertheless, additional work in other molecularly defined cohorts with larger numbers may be needed to confirm our results.

## 5. Conclusions

Finally, beyond the prognostic impact, we were able to closely correlate p16 immunohistochemical expression and *CDKN2A* homozygous deletion. As expected, p16 overexpression was never associated with *CDKN2A* homozygous deletion. An absence of p16 expression was only related to *CDKN2A* homozygous deletion in two-thirds of cases, suggesting the influence of another silencing phenomenon in the remaining cases, perhaps relating to transcriptional or post-transcriptional rearrangements. Regarding the pattern of focal expression, a 5% cutoff of p16-positivity detected all *CDKN2A* homozygous deletions in our cohort and was associated with a very high sensitivity and negative predictive value for detecting a *CDKN2A* homozygous deletion with this IHC method. Furthermore, the similarities between the survival curves using both techniques suggest that p16 immunohistochemical expression and *CDKN2A* deletion evaluated by FISH assay give similar information regarding prognostic implications. So, IHC might be proposed as a pertinent surrogate test to evaluate *CDKN2A* status when using our cutoff, and it will provide adequate prognostic data in each glioma subset. 

## Figures and Tables

**Figure 1 cancers-15-01512-f001:**
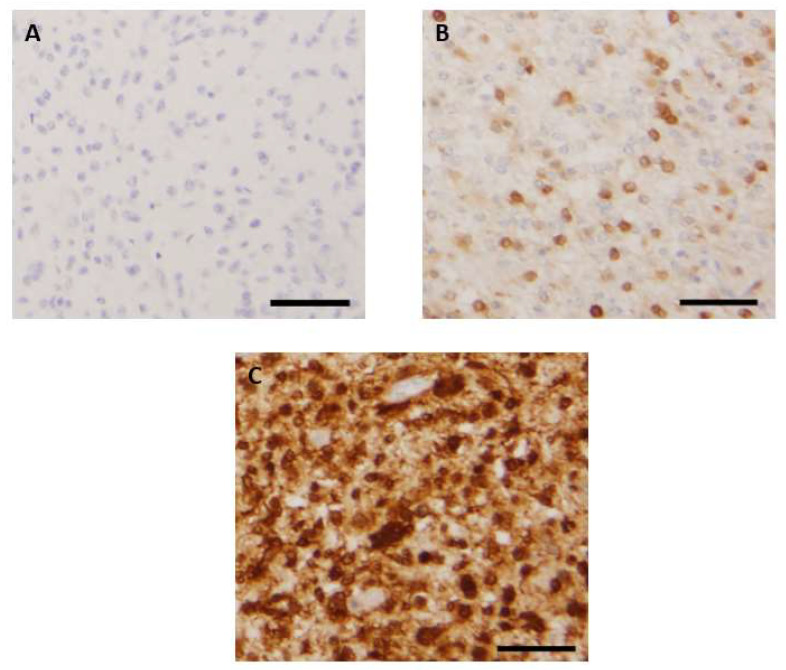
Representative images of patterns of p16 immunohistochemical staining: absence of expression (**A**), focal expression in a variable amount of tumor cells (**B**), and intense and diffuse expression in a large majority of tumor cells (**C**). Scale bar is 50 µm for Figure 1A–C.

**Figure 2 cancers-15-01512-f002:**
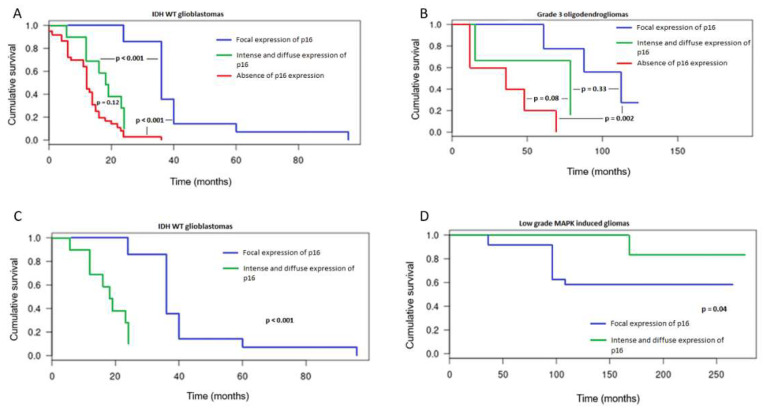
Kaplan–Meier survival curves according to the pattern of p16 expression in *IDH*-wt glioblastomas (**A**) and *IDH* mutant 1p/19q codeleted oligodendrogliomas (grade 3) (**B**). Comparison of Kaplan–Meier survival curves according to the pattern of p16 expression in *IDH*-wt glioblastomas and MAPK-induced gliomas (**C**,**D**). *p*-value under 0.05 was considered statistically significant.

**Figure 3 cancers-15-01512-f003:**
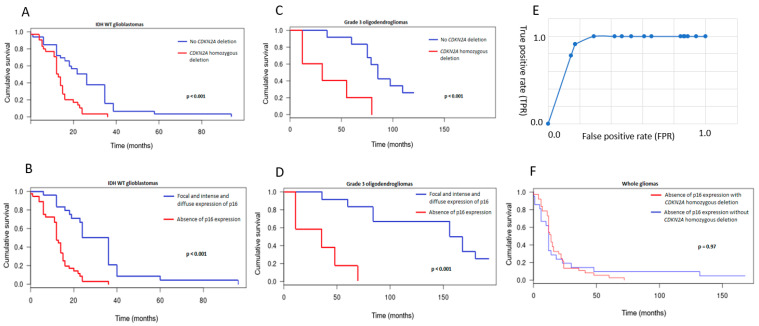
Kaplan–Meier survival curves according to *CDKN2A* homozygous deletion in *IDH*-wt glioblastomas (**A**). Kaplan–Meier survival curves according to the pattern of p16 expression in *IDH*-wt glioblastomas (**B**). Kaplan–Meier survival curves according to *CDKN2A* homozygous deletion in oligodendrogliomas (grade 3) (**C**). Kaplan–Meier survival curves according to the pattern of p16 expression in oligodendrogliomas (grade 3) (**D**). ROC analysis plot of the performance of p16 IHC in detecting *CDKN2A* homozygous deletion (**E**). Kaplan–Meier survival curves according to *CDKN2A* deletion in the p16 negative groups (**F**). *p*-value under 0.05 was considered statistically significant.

**Figure 4 cancers-15-01512-f004:**
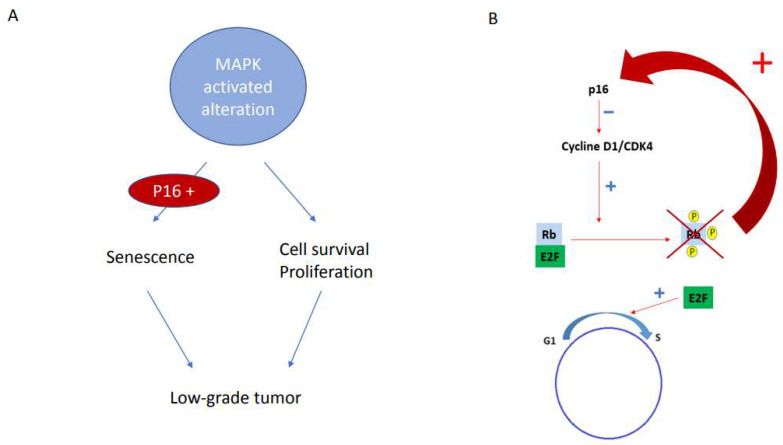
Schematic representation of the Oncogene-induced Senescence (linked to the activation of the MAPK pathway) leading to low-grade tumors (**A**). Schematic representation of p16 overexpression in high-grade gliomas linked to the positive feedback related to RB1 deletion (**B**).

**Table 1 cancers-15-01512-t001:** Clinical characteristics of the cohort.

Characteristics	Number (%) (n = 173)	Number of Deceased Patients (%)
Median age (years)	36.5 (1–79)	
Sex		
Male	95 (55)	
Female	78 (45)	
Adult-type diffuse gliomas		
*IDH*-mutant astrocytoma, grade 2	4 (2)	1 (25)
*IDH*-mutant astrocytoma, grade 3	5 (3)	4 (80)
*IDH*-mutant astrocytoma, grade 4	6 (3)	6 (100)
*IDH* mutant and 1p/19q codeleted oligodendroglioma, grade 2	10 (6)	3 (30)
*IDH* mutant and 1p/19q codeleted oligodendroglioma, grade 3	17 (10)	16 (94)
*IDH-*wt glioblastoma, grade 4	63 (36)	63 (100)
Circumscribed gliomas		
Pilocytic astrocytoma	36 (21)	3 (8)
*Fusion KIAA1549 :: BRAF*	27	
*BRAF V600E mutation*	1	
*NF1 mutation*	1	
*FGFR1 duplication*	1	
*NOS*	6	
Pleomorphic xanthoastrocytoma with anaplasia	1 (1)	0 (0)
Glioneuronal tumors		
Ganglioglioma	10 (6)	0 (0)
*Fusion KIAA1549 :: BRAF*	2	
*BRAF V600E mutation*	8	
Other low-grade pediatric gliomas		
Subependymal giant-cell astrocytoma	2 (1)	0 (0)
Angiocentric glioma with MYBL1 alteration	1 (1)	0 (0)
High-grade pediatric-type diffuse gliomas		
Diffuse midline glioma H3K27-altered	9 (5)	9 (100)
High-grade diffuse pediatric glioma, H3 and *IDH-*wt	8 (4)	8 (100)
Diffuse hemispheric glioma H3 G34-mutant	1 (1)	1 (100)
Deceased patients	114 (66)	
Operated tumor recurrence	33 (19)	
Surgical specimens		
Biopsy	10 (6)	
Resection	163 (94)	

**Table 2 cancers-15-01512-t002:** Patterns of p16 expression in the whole cohort.

Diagnostic	Number of Cases	Absence of p16 Expression (Nb)	Focal p16 Expression (Nb)	P16 Overexpression (Nb)
*IDH*-mutant astrocytoma, grade 2	4	0	4	0
*IDH*-mutant astrocytoma, grade 3	5	2	2	1
*IDH*-mutant astrocytoma, grade 4	6	2	3	1
*IDH*-mutant, 1p/19q codeleted oligodendroglioma, grade 2	10	0	10	0
*IDH*-mutant, 1p/19q codeleted oligodendroglioma, grade 3	17	5	10	2
*IDH*-wt glioblastoma, grade 4	63	36	17	10
Pilocytic Astrocytoma, grade 1	36	1	30	5
Anaplastic PXA, grade 3	1	1	0	0
Ganglioglioma, grade 1	10	0	5	5
Other low grade gliomas, grade 1	3	0	3	0
Diffuse midline glioma, H3K27-altered, grade 4	9	6	1	2
Diffuse pediatric-type glioma, H3 and IDH wt, grade 4	8	3	3	2
Diffuse hemispheric glioma, H3-G34-mutant, grade 4	1	0	1	0
Total	173	56	89	28

## Data Availability

The entire and detailed immunohistochemistry data and FISH data are available upon request to the corresponding author.

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
