# Peer review of "p16 Immunohistochemical Expression as a Surrogate Assessment of CDKN2A Alteration in Gliomas Leading to Prognostic Significances"

_cancers, 2023, doi:10.3390/cancers15051512_

Round 1

Reviewer 1 Report

Geyer at all investigated the use of a p16 immunohistochemical stain as a surrogate for a CDKN2A alteration in gliomas. This particular mutation is a diagnostic requirement for a WHO grade 4 tumor.

Abstract: The abstract accurately summarizes the publication is written.

Introduction: The introduction describes the background for testing of CDKN2A, and its diagnostic and prognostic importance. The hypothesis was that p16 expression, which has a commonly used immunohistochemistry stain, can be used as an easily performed surrogate for detection of CDKN2A alterations.

Materials and methods: This is a retrospective analysis of 173 gliomas of various grades diagnosed between 1989 and 2012 at a single institution. Specimens were reassessed to conform to the 2021 most recent WHO classification. The description of data acquisition including clinical characteristics, pathology evaluation, immunohistochemical and CDKN2A evaluations and statistical analysis were adequately detailed.

Results: Results provided a comprehensive summary of the results found. All associated tables and graphs are appropriate with one exception: Figure 1 consists of three figures and a table. The table should be separate. This particular table also reiterates some information that is present in the narrative, but is appropriate as it organizes important information with regard to tumor types and p16 expression.

Discussion: The discussion adequately describes the detected correlation between p16 overexpression, CDKN2A results, and Rb1 deletions, and describes which tumor types would benefit from p16 immunohistochemistry rather than CDKN2A testing via FISH or other methods. The authors adequately describe parameters in which p16 immunohistochemistry may be used in lieu of CDKN2A results.

Conclusions: The conclusion section adequately discusses the recommendations for further investigation regarding the use of p16 immunohistochemical expression when other techniques to detect a CDKN2A deletion cannot be performed.

Figures and tables: see results section for concerns regarding figure 1, which should be split into a figure with the three representative pathology slides and a separate table regarding the presence or absence of p16 expression in various tumor types.

References: All references are appropriate and relevant to the research.

Author Response

We thank the reviewer for his report.

As requested, figure 1 is now split into a supplementary table and a figure.

Moreover, the manuscript has been reviewed by a native english speaker and minor spell checks were carefuly corrected 

Reviewer 2 Report

General comments

This is an interesting study that has day to day relevance for management since it suggests a straightforward approach to investigating a relevant prognostic biomarker. The data are novel and will be of interest to a Neuro oncology readership.

My only general criticism is that the standard of English is a little variable and needs to be improved throughout. I would suggest a review by an English speaker. There are also a few typos eg RB11 in line 201.

Specific comments

The inclusion of all types of glioma (adult and paediatric)should be justified a bit more clearly. Obviously this introduces significant heterogeneity which reduces the relevance of the outcome data across the whole data set. The total numbers (173) over a 23 year period also seems quite a small data set. The authors should explain what % of the patients that went through the service this represents and what is the reason for excluding some cases.

In the results section there is no mention of multi-variate analysis to confirm the independent effect of CDKN2A loss, which should be included.

The discussion should also include some recommendation for additional work, for example undertaking the same work in molecularly defined cohorts with larger numbers to validate this approach.

Author Response

We are very grateful for your report allowing us to improve our manuscript.

The manuscript has been reviewed by an native english  speaker and typos were carefully corrected included RB11 in line 201.

Regarding the inclusion of cases:

  • First, a typo was unfortunately done in the material and method regarding the beginning of inclusion. In fact, the period of inclusion is from 1999 to 2012 and not 1989 to 2012. This is corrected in the manuscript (line 94).
  • Second, most of the cases were from 2005 to 2012 (adult type diffuse gliomas). We only extended the period for rarer cases, especially BRAF alterated gliomas. 
  • Third, biopsies were excluded due to limited FFPE allowing the realization of TMA and additionnal molecular test
  • Four, cases without inform consent and clinical follow-up were excluded

All this cases represent about 35% of all the patient with gliomas operated in the neurosurgery department of Strasbourg University Hospital.

Material and method is now reviewed.

Regarding the results, multivariate analyses has been added.

Finally, discussion now includes recommandations for additionnal work.